# Future Perspective and Technological Innovation in Cheese Making Using Artichoke (*Cynara scolymus*) as Vegetable Rennet: A Review

**DOI:** 10.3390/foods12163032

**Published:** 2023-08-12

**Authors:** Michael Steven Bravo Bolívar, Federica Pasini, Silvia Marzocchi, Cesare Ravagli, Paola Tedeschi

**Affiliations:** 1Department of Agricultural and Food Sciences and Technologies, University of Bologna, Piazza Goidanich 60, 47521 Cesena, Italy; michael.bravobolivar@studio.unibo.it (M.S.B.B.); federica.pasini5@unibo.it (F.P.); cesare.ravagli2@unibo.it (C.R.); 2Interdepartmental Centre of Industrial Agri-Food Research (CIRI Agroalimentare), University of Bologna, Via Quinto Bucci 336, 47521 Cesena, Italy; 3Department of Chemical, Pharmaceutical and Agricultural Sciences, University of Ferrara, Via Luigi Borsari 46, 44121 Ferrara, Italy; paola.tedeschi@unife.it

**Keywords:** *Cynara scolymus*, vegetable rennet, coagulating capacity, protease enzymes, *Cynara cardunculus*

## Abstract

Milk coagulation is a process used for the formulation of different dairy products such as cheese. In this process, milk undergoes changes in its chemical stability thanks to acidification or enzymatic reactions. Traditionally, milk coagulation has been carried out with rennet of animal origin, but recently, the research of new types of rennet such as microbial rennet and vegetable rennet has increased. This study aims to present an organized review of the most relevant information on lactic coagulation, its relationship with vegetable rennets, and the importance of the botanical genus *Cynara* in the extraction of vegetable rennets, focusing on the coagulant potential of artichoke (*Cynara scolymus*). We conducted this literature review and found that lactic coagulation and vegetable rennets are linked through the enzymatic activity of the latter. The results of the main studies demonstrated a strong relationship between vegetable rennets and protease enzymes as well as the presence of these enzymes in extracts of cardoon (*Cynara scolymus*) and artichoke (*Cynara scolymus*). In addition, studies highlight the presence of thistle extracts in artisanal cheese preparations in the Iberian Peninsula. Based on the results of the studies, a comparison between cheeses made with vegetable rennet and those made with traditional rennet was also carried out. Although the results show that the use of vegetable rennet in the manufacture of cheese can confer undesirable characteristics, the use of extracts from *Cynara* plants demonstrates that vegetable rennets have an industrial potential, especially the one obtained from artichoke (*Cynara scolymus*) due to its high availability. Nevertheless, specific studies are required for a better understanding and application of this rennet.

## 1. Introduction

The milk coagulation process takes place by acidification or by enzymatic action on phenylalanine and methionine 105 and 106, both present in the milk к-casein. This causes the milk to destabilise and separate into two parts: one consisting mainly of water and the other consisting mainly of fat and proteins present in the milk, constituting a new three-dimensional mass. The latter forms the basis of cheeses and other dairy products [1,2,3].

Ben Amira in his review explained that initially animal rennet was responsible for milk coagulation, but the sudden increase in the population led to the need of new rennet options, such as that of non-animal origin. At the beginning, the most popular ones were essentially of microbial or fungal origin [1]. In addition to the increasing population, the rising prices of animal rennets, the increasing vegetarian population, and religious customs aroused the interest in researching other types of rennets, such as vegetable and microbial ones [2]. 

As Silva¨ [2] and her teamdescribed, the new generation of research on enzymes capable of coagulating milk is largely focused on the formation of plant-derived rennets due to their high availability and their generally simpler extraction and purification processes compared to other rennets. Her work highlighted the use of various types of plants, such as *Asteraceae*. In this category, wild thistles, cultivated cardoons (*Cynara carduculus*), and artichoke (*Cynara scolymus*) stand out. All three plants may be viable candidates for the extraction of vegetable rennet. 

Historically, artichoke has been part of the daily life of certain cultures, especially in Europe. About 40% of the world’s total production is carried out in this continent, as artichoke is part of the Mediterranean and other European diets. For this reason, for several years Europe, has reached a production of more than 620,000 tonnes per year [4]. 

As explained by Lattanzio’s research group, only a part of the artichoke is currently utilised (15–20%) since the rest is considered a waste within the agricultural industry [5]. This is a high percentage of waste compared to other edible plants, such as lettuce, which can reach about 70–75% of edible biomass if it is produced through hydroponic processes [6]. Due to this high percentage of waste in artichoke production, the discarded biomass is a possible source of value-added products. Lattanzio’s research showed that artichoke leaves and stems can be significant sources of polyphenolic compounds, caffeoylquinic acids, flavonoids, as well as inulin, a polymeric molecule of fructose that is of high interest due to its potential as a sweetener and as a source of fermentation [5]. 

In addition to the possible use as a source of value-added molecules, the studies in *Asteraceae* plants define artichoke as a potential of proteases and enzymes responsible for milk coagulation. This was demonstrated by the research of Llorente [7] and her research team, which identified artichoke as one of the best candidates for vegetable rennet synthesis and thus for cheese manufacture [7]. Therefore, the objective of this study is to provide an insight into the relationship between the milk coagulation process and vegetable rennets, highlighting the role of the botanical genus *Cynara* in the manufacture of vegetable rennet, with special emphasis on the possible use of artichoke (*Cynara scolymus*) as a vegetable rennet due to its wide availability in the Mediterranean region. 

## 2. Materials and Methods

This section delineates the approach employed to conduct an exhaustive review on vegetable rennets and their milk coagulation activity, with a particular focus on artichoke rennet. The steps to identify, select, and analyse the necessary information are hereinafter described.

### 2.1. Identification of Relevant Articles

The first step of our literature study entailed a comprehensive search of prominent scientific databases, including *Elsevier*, *ScienceDirect*, *Springer*, *Google Scholar*, official websites of the European Union, and databases of PDO products, to find pertinent publications on vegetable rennets and their coagulation activity. To refine our search, we utilised targeted keywords such as “vegetable rennets”, “milk clouting activity”, “enzyme proteases from artichoke”, and “*Cynara carduculus*” and “*Cynara scolymus* as rennet for cheese making”. The search was limited to a period from 1990 to 2022 in all databases. Note that this period does not include the access date of official websites, and the Curtis’s botanical magazine of 1930 was used only as a botanical reference to create the figures.

### 2.2. Study Selection

Mendeley reference manager software, version 2.80.1, was used to manage the bibliography in an orderly fashion.

After comparing and eliminating duplicate articles, the remaining articles were screened by title and abstract. After screening, a final selection was made on the basis of the inclusion criteria. 

The inclusion criteria are the following: (1) The articles had to include these topics: relationship between the phenomenon of lactic coagulation, vegetable rennets, and the artichoke (*Cynara scolymus*); description of results in the use of vegetable rennets; and comparison between different types of rennets. (2) The papers had to be published in peer-reviewed scientific journals. (3) The papers had to be written in English. We included only one source that does not meet these characteristics, which is a study conducted at the University of Extremadura related to the manufacture of PDO cheeses and written in Spanish language. 

### 2.3. Data Extraction and Analysis

After summary screening, pertinent data were extracted from the selected articles focusing on both the data presented and the bibliography included in the discussion of results, ensuring the maintenance of the relationship between the contents found and the objective of this work.

We compared the data extracted from the selected articles to find the key points of connection between them and to develop the writing strategy for this work.

## 3. Key Factors of Milk Coagulation

As previously mentioned, the main stage of cheese making depends to a significant extent on the enzymatic reactions that are generated during the coagulation process, but they are not the only ones responsible for optimal development of milk coagulation. Different factors are also involved, such as the origin of the milk, its chemical characteristics, as well as temperature, pH, and specific operating conditions [3,8].

### 3.1. Chemical Characteristics of Milk 

As defined in international food standards, milk is a mammary secretion intended for consumption, which may have unique characteristics depending on its origin. Regardless of its origin, it has some general characteristics: the presence of lactose, micronutrients, vitamins in varying percentages, a high percentage of water, the presence of fats (at least 3%), and a variable percentage of protein, among which various types of casein stand out [9]. However, as explained later, the characteristics of milk depend on various aspects that go beyond the type of animal that produces milk. For instance, Soares [10] and her working group claimed that the composition of milk is largely influenced by animals’ diets [10]. However, general characteristics concerning the components of milk according to origin exist (Table 1).

### 3.2. Casein in Milk 

Casein is an essential component of milk: it accounts for at least 80% of milk protein, which may represent about 2.5% of the weight of milk. There are several types of caseins (α_s1_, α_s2_, and β- and к-casein) available in milk, which are present in different proportions depending on the type of milk [19]. They are important for the stability they confer to milk through the formation of casein micelles.

An important characteristic of micelles is that they are highly hydrated molecules, with a ratio of about 3.5 kg of water per 1 kg of protein, representing a significant portion of the total milk volume. Micelles can be destabilised by pH changes or enzymatic action, resulting in the separation of milk and the formation of other milk products [19,20,21]. 

Milk micelles are mainly characterised by their complex structure featuring к-casein in the outer layer of the micelles, which is not as thick as would be necessary to prevent the passage of certain molecules into the micelle. As explained in Anema’s work, when milk is heated to moderate temperatures (45–65 °C), the outer layer of the micelles allows the β-casein to escape, and when the temperature is lowered, it returns to the interior of the micelle. The explication of this behaviour is reinforced by the structural model proposed by Walstra and Jenness that defines the micelle as a conformation of smaller subunits located on the inside, which are formed by the α_s_ and β-casein, and another group formed by α_s_ and к-casein that is located on the outside due to the interaction of the к-casein with sugars. This forms a more hydrophilic outer layer and a more hydrophobic inner layer (Figure 1) [19,21,22,23,24]. 

The coagulation of milk is mainly due to two processes: the acidification of milk and the proteolytic activity of rennet, the latter being of major industrial interest. During this process, the casein micelles are destabilised, leading to the creation of other products [25]. As explained in the work of Britten [3] and his team, coagulation takes place in two main phases.

The first phase of coagulation begins with the enzymatic hydrolysis of к-casein, which, as mentioned earlier in this work, is the casein that occupies the outer part of the micelles; this phenomenon is caused by the hydrophilia of the terminal region of к-casein. As explained in Horne’s work, this characteristic of к-casein allows the micelles to be more stable, but the hydrophilic terminus allows chymosin and other rennet enzymes to break the Phe105–Met106 bonds, resulting in the release of a casein peptide (caseinomacropeptide) during the serum phase of the milk [3,26]. Kethireddipalli [27] and his team explained that after acid precipitation of the proteins, it is possible to detect the hydrolysis of к-casein by detecting the formation of the aforementioned peptide by RP-HPLC, which allows to monitor the process and to better establish when the second phase of coagulation starts [27]. The second phase consists of a gel formation caused by the aggregation of destabilised micelles [26]. This occurs after the hydrolysis of most of the к-casein, caused by hydrophobic interactions that occasion the formation of calcium bridges. These bridges form a highly porous, three-dimensional network where serum and fat globules can be trapped [3]. The aggregation phenomenon of this stage can be affected by factors such as pH and process temperature [3]. However, as explained in several texts, one of the most important factors of this phase and one essential for the characteristics of some types of cheese is the presence of ionic calcium and its interaction since it is especially useful to accelerate the aggregation and gelation phenomenon, providing greater firmness of the resulting gel [28,29]. 

## 4. Use of Vegetable Rennet for Cheese Making

As already mentioned, there is a growing interest in researching new types of rennet, such as those of bacterial origin and those from plant enzymes [1]. As established in previous works, plants have a high enzymatic activity, and sometimes, they can be suitable for the manufacture of cheese and other dairy products due to the presence of proteases and other enzymes [30,31,32]. 

*Asteraceae* plants form a group with a significant potential for the development of milk coagulation. Studies focused on this broad family prove this potential: one of its characteristics is the presence of several groups of enzymes where different types of proteases can be found [33]. 

Although, as Lo Piero explained in his work, the use of some vegetable rennets is industrially limited by factors such as the proteolytic activity of some vegetable extracts, it can be very high and confer some organoleptic defects to the cheeses made with these rennets [34]. However, the development of the use of vegetable rennet is not recent since typical cheeses in countries such as Spain and Portugal have been made with this type of extracts for a long time [1,35]. This indicates that cheese production can be influenced by the type of enzyme extracted from the plant, the origin of the plant, and the method of obtaining the extract, among other factors [36].

### Main Enzymes in Vegetable Rennets Responsible for Coagulation

During the life span of plants, they use various enzymes to fulfil certain functions, such as proteases, which are present in processes related to germination as well as in cell death reactions [1,30]. 

As previously mentioned, the coagulation of milk by enzymatic means is usually related to protease enzymes. Regarding vegetable extracts, enzymes derived from aspartate, serine, and cysteine proteases are capable of performing coagulating functions in milk, according to reports (Table 2) [1,2,30]. 

Aspartic proteases are more active at a relatively low pH range (3–5) [43] and are very selective in some types of peptide bonds in hydrophobic amino acid residues. Therefore, they are of great importance since these types of bonds are present in molecules such as casein [30]. Research has demonstrated the presence of this type of protease in a considerable number of plant species such as artichoke (*Cynara scolymus*) [7,38] and various types of cardoons such as the marian (*Silybum marianum* L. Gaertn.) [30,44] and cardoon flower (*Cynara cardunculus*) [45], among other species [30]. Artichoke and cardoon flower are the most studied since the latter is traditionally used in the manufacture of cheese in some parts of Spain [35]. 

Cysteine proteases have an important value in the food and pharmaceutical industry because they can be used for a wide range of activities at various temperatures and pH, which increases its versatility and industrial utility [46]. It was found that their optimal activity temperature is 80 °C, but this temperature is not implemented in dairy processes [47]. In addition, these proteases are present in various types of tissues of multiple plants, which simplifies their availability and use [30]. It has been proven that various types of enzymes of this group of proteases can be present in plants, such as latex (*Ficus racemosa*), which has proven their potential use in the dairy industry due to their ability to break casein bonds. Moreover, it was found that proteases with coagulation capacity in milk are present in kiwis (*Actinidia chinensis*) as well as in ginger (*Zingiber officinale*) [30,31]. It is also worth noting that cysteine-type proteases belong to 108 different families [48], which may explain their presence in multiples plants species. 

Serine proteases can perform nucleophilic attacks on the carbonyl groups of the peptide bond of various types of substrates, forming enzyme/substrate complexes [31,49]. These enzymes are present in almost all plant tissues and in a wide variety of plants, including legumes, trees, and herbs. Their great variety has made their extraction from stems, seeds, flowers, some leaves, and roots possible [30]. This type of proteases has been used to coagulate milk. Several studies have reported both extraction and purification of some thermostable enzymes, for example, the serine protease from fennel (*Foeniculum vulgare*), which is stable in a determined pH range (6 to 7.5) and temperature range (40 °C to 60 °C) [50]. Other examples are religiosin in different variants (a, b, and c), extracted from some serine proteases from latex (*Euphorbia neriifolia*) [30], and also the ones capable of coagulating milk extracted from lettuce (*Lactuca sativa*) and purified [34].

## 5. Role of *Cynara* Genus 

As can be observed in the table, there are several plants with milk-coagulating capacity (Table 2), but as mentioned in this work, in a large number of cases, this coagulating capacity does not generate high-quality cheeses, although there are suitable plant candidates for the formulation of cheeses with excellent organoleptic characteristics [34].

As it has been reported in previous works by multiple authors and as highlighted in this work, the plants of the *Cynara* genus have an important potential for the formulation of high-quality cheeses to the point that there are several cheeses with the category of protected designation of origin (PDO), as reported in (Table 3). Therefore, some types of thistle and artichoke are the most studied candidates and may be optimal for the industrial cheese manufacture [7,35,38,45].

### 5.1. Importance of Cardoon (Cynara cardunculus) in the Production of Vegetable Rennet

Cardoon (Figure 2) is one of the most studied plants for the possible generation of vegetable rennets at industrial level, and this is due to the fact that some PDO dairy products in Spain and Portugal are made using extracts of different types of thistles (Table 3). This proves its current widespread use and its possible presence at the industrial level [35,53].

The widespread use of cardoon as a milk coagulant has generated high interest due to its high availability throughout the Mediterranean, although, as discussed in this paper and by Jacob [56] and his group work, in some cases, the cardoon has a high proteolytic activity that can promote the loss of protein and fatty material in the formed gel [56]. This impediment does not seem to affect the organoleptic characteristics of some PDO cheeses from the Iberian Peninsula (Table 3), perhaps because most of them are made with sheep’s milk, which contains more protein than cow’s milk (Table 1). This creates a compact gel, although the proteolytic activity promotes losses of protein material in the whey, resulting in a homogeneous final product and also a whey rich in protein. In some cases, these cheeses are made with different types of milk, which may influence the protein content of the gel and whey [51,52]. Additionally, this loss of yield becomes an advantage when protein-rich whey is used to form by-products, as in Portugal, where protein-rich whey is used to manufacture whey cheese (Requeijão), recovering protein losses [1]. 

### 5.2. Thistle Extract 

The traditional use of cardoon as rennet involves the extraction of its enzymatic potential, which is generally performed on the flower (Figure 2C) and its dried pistils (Figure 2E) using weak acids or just water [45,53,57]. 

The extract is exclusively obtained from the mature flower and pistils (Figure 2C,E) because, as explained in the work of Cordeiro [58,59], it is presumed that this protease enzyme activity in the mature flower is due to a species protection system against some viruses or microorganisms at the time of pollination [60].

As explained in García’s work and Ordiales’s work [57,61], cardoon extract presents a great variety of aspartic proteinase enzymes, which belong to the group of endopeptidase enzymes. In this group, pepsin and chymosin stand out, making clear the coagulating potential of the extract from this cardoon species. 

Cardoon extract (*Cynara cardunculus*) is characterized by significant amounts of cynarases or cyprosins, two groups of proteinases that have been shown to be determinant in the coagulating capacity of cardoon extract. In particular, cardosins A and B and cyprosins 1,2,3 have been identified [58,59,62]. 

As shown in the work of Pino and his group, it is possible to compare the coagulating activity of cardoon extract (*Cynara cardunculus*) with the activity of traditional rennets because the enzymes cynarases and cyprosins are able to affect к-casein [63], promoting the first complete phase of milk coagulation.

The comparison [63] showed that in the manufacture of cheeses with a ripening time of less than 5 months, there were no significant differences between the coagulating power of traditional rennet and cardoon extract. In addition, characteristics such as pH, humidity, and NaCl concentration presented similar values according to the ripening time of the two types of rennet. However, it was also found that soluble nitrogen was considerably higher in cheeses made with cardoon extract. This phenomenon was also evidenced by authors who compared cheese manufacture made with cardoon extract and other coagulants [64,65].

## 6. Artichoke as a Source of Clotting Enzymes

The artichoke (*Cynara scolymus*) (Figure 3) is of great importance in the Mediterranean region due to its wide consumption, with an annual production of 620,000 tonnes in Europe [4], which makes clear its importance at the agricultural level. However, its relevance is growing in other industrial perspectives due to its possible use in other areas of the food industry.

As demonstrated by Llorente [7] and her team in Gouda cheese production, artichoke has already been used to produce cheese. A comparison of gel-formation times between standard rennet and artichoke extract showed no significant differences since both formed rennets with similar characteristics. This type of research and the important role of plants such as the cardoon (*Cynara cardunculus*) and others of the *Cynara* genus presented in this work strengthen the perspective of using artichoke as a vegetable rennet. 

### 6.1. Enzyme Activity of Artichoke (Cynara scolymus)

Artichoke (*Cynara scolymus*) presents an important coagulant activity due to the presence of protease enzymes in several of its tissues, as in other plants of the *Cynara* genus. The work of Llorente [38] and her research group revealed that artichoke presents a relevant enzymatic activity in several organs [38]. It was shown that it is possible to extract different types of enzymes from roots, young leaves, and adult leaves of mature and immature artichoke flowers. 

Like other plants, artichoke contains a lot of bioactive compounds, and the total antioxidant capacity of artichoke flower head is one of the highest reported for vegetables [67]. Artichoke displays a wide range of stress tolerance thanks to its milk-clotting enzymes and an important antioxidant system composed by antioxidant enzymes such as catalase peroxidase and ascorbate peroxidase. In some plants (e.g., *Ficus carica*, *Carica papaya*, and *Calm viscera*) enzyme clotting activity has been identified, but all these enzymes were found unsuitable since they produce bitter cheese due to their strong proteolytic activity. An exception is represented by artichoke flower extracts, which contain multiple molecules of aspartic proteases [68,69]. 

The high enzymatic activity of artichoke places it in a privileged position compared to plants of the same genus, such as the cardoon, because its enzymatic activity in the mature flower as well as in other organs may offer better industrial advantages due to its high availability and the growing interest in artichoke among consumers [5,70]. Nevertheless, as Llorente also explained in her research [38], even though artichoke presents enzymatic activity in several organs, as represented in (Figure 3), not all of them are suitable for lactic coagulation, and milk coagulation depends on a specific enzymatic action on the structure of casein micelles. However, Llorente’s work also demonstrated that several artichoke organs that exhibit enzymatic activity (immature flowers, mature leaves, and mature flowers) also have a specific milk-coagulation ability. The mature flower extract is the one with the highest activity.

#### 6.1.1. Artichoke and Its Coagulant Activity on Milk 

As previously mentioned, various artichoke organs presents enzymatic activity in milk, but research has focused above all on extracts from mature artichoke flower due to its greater coagulant activity and probably due to its possible comparison with the extract of the cardoon flower, which has been more investigated and utilised [2,5,7,45]. This research reveals that artichoke and cardoon flower extracts have a coagulating capacity similar and comparable to the more commonly used animal or microbial rennets. 

By observing these specific coagulation characteristics on milk, it was determined that artichoke flower extract presents several protease enzymes, such as cardosin A and B [70]. However, other works have shown that artichoke extracts can have several fractions with coagulant capacity in milk. Cynarase isolated and extracted from *Cynara scolymus* has been described in many studies [39,40,71] as an aspartic protease with two very important fractions for the catalytic activity: aspartic acid residues in the catalytic site and other aspartic preoteases that were most active at acidic pH and showed high preferential specificity for cleavage sites where the peptide bonded between hydrophobic amino acid residues [72].

The work of Chazarra demonstrated the presence of cynarases A, B, and C [39] glycoproteins containing N-linked high-mannose-type glycans [40]. Moreover, the enzymes of coagulant activity may be decreased by purification of artichoke extract compared to the crude extract of cynarases A and C, while cynarase B increased its milk-clotting activity.

Furthermore, subsequent studies on the purification of the artichoke flower extract resulted in the formation of five different fractions with coagulant potential in milk (a, b, c, d, and e). In these fractions, as Llorente explained, the cynarases cannot be identified because detailed studies on each fraction have not yet been carried out [4,7,34,36]. Nevertheless, based on the reports of other the authors mentioned above, such as Chazarra, and according to the similarity with cardoon, it is likely that they could be fractions belonging to this group of enzymes.

#### 6.1.2. Artichoke Coagulant Extract

Artichoke presents protease enzymatic activity in various organs, and usually, its enzymatic potential is extracted with the use of water or weak acids, always generating an aqueous extract [38,61]. However, as explained in the work of Chazarra, the coagulating activity of artichoke extract is affected by various factors such as pH since the extracts formed in a pH range between 3 and 7 are functional and have specificity for milk, but lower pH values report higher performance in coagulation [39]. However, it is important to emphasize that, as mentioned in this work, the pH of the medium influences milk coagulation since only a decrease in pH can promote the breaking of the emulsion, so it is not clear whether the pH of the artichoke extract itself favours the milk-coagulation process. On the other hand, as also explained by Chazarra and other authors such as Bueno-Gavilá, the difference between an extract obtained in a medium with an acid pH and another obtained at a pH closer to 7 is minimal [39,73], which demonstrates the high effectiveness of artichoke extract in promoting the aforementioned first phase of milk coagulation.

Using and obtaining a coagulating extract in aqueous form is widespread due to previous reports and also due to the use of thistle rennet in some commercial cheeses, as reported in Table 3. However, artichoke presents different alternatives already reported with favourable perspectives regarding its effectiveness as rennet. Heba-Allah reported the use of agricultural residues from artichoke cultivation, which are first transformed into a powder, and then, the powder is taken to aqueous extraction or with various types of buffers. The latter extract is the one used as rennet [71]. Very different alternatives are presented in the study by El-Kholy [74] and his team, who reported the use of raw stylets and stigmas of artichoke flower (Figure 3D) for cheese making and presented a comparison between the coagulating potential of raw stylets and stigmas, an aqueous extract of artichoke flower, and an animal rennet standard. In his study, the stylets’ and stigmas’ quantity in relation to the volume of milk used was also evaluated. The result was a cheese widely accepted at the sensory level and presenting differences related to nitrogen content. These differences, previously mentioned in this work, can be attributed in general to the use of vegetable rennet in the manufacture of certain types of cheese.

### 6.2. Future Prospect for Artichoke Use

As already explained in this work, the artichoke is an important part of the Mediterranean diet and has been important for the agriculture of this region. For this relevance, Europe is the largest producer in the world, and the Mediterranean is the largest producer region in Europe [4]. Taking into account this large production at the European level and knowing that the percentage of post-consumption biomass is high, authors such as Lattanzio [5] suggested the possible use of artichoke as a source of other value-added raw materials. As artichoke cultivation has a large commercial and scientific diffusion, interest in artichoke and in the biomass formed during its cultivation may increase.

As was exposed by several authors [2,7,38,70,75], much of the artichoke enzymatic activity is given by protease enzymes, positioning the artichoke as a possible source of these enzymes. Moreover, as Llorente explained [38], the artichoke presents this type of activity in several of its organs, unlike similar plants such as the cardoon, where it is only possible to find this enzymatic activity of proteases in the flower [58,59], positioning artichoke above other plants as a good source of this type of enzyme. In addition, in the future, artichoke could be used to carry out studies to improve the extraction and purification of coagulating enzymes and evaluate their possible use at industrial level.

Considering that artichoke (*Cynara scolymus*) and cardoon (*Cynara cardunculus*) have botanical and chemical similarities, similar protease enzyme activity, and the same type of protease enzymes [45,61,75] and that cardoon is currently used as rennet in the manufacture of PDO cheeses, it is possible to establish a possible use of artichoke flower extracts as rennets. As shown in this work and as demonstrated by several studies [63,65,71,74], this is also possible due to the minimal variations between the organoleptic characteristics of a cheese made with cardoon or artichoke rennet and one made with standard animal rennet. However, it should be noted that, as mentioned in this paper and in the work of other authors [63,64,74], if there are chemical differences (such as in nitrogen content) between cheeses made with animal rennet and vegetable rennet, they may affect the production of some types of cheese. For this reason, studies on artichoke rennet used in cheeses with long maturation or with different types of milk are necessary to obtain more data for comparison with animal rennets or other vegetable rennets.

### 6.3. Possible Impediments to the Use of Artichoke as a Vegetable Rennet

As previously stated, and as Lo Piero explained in his paper [34], in some cases, the use of vegetable rennets can confer undesirable flavours due to their high proteolytic activity, and this can be an impediment in the use of artichoke extracts as rennet in the manufacture of certain types of cheeses. This poses a research challenge. It is important to choose the right organ to extract the coagulant potential since they possess enzymatic activity and specificity for milk-coagulating differences [38]. This unique characteristic of artichoke may lead to further research focused on the use of organs with milk specificity but lower enzymatic activity, elucidating how to reduce the proteolytic activity of vegetable rennets and leading to a future development of a more detailed method to control the coagulation of milk with vegetable rennets.

Another possible impediment to the use of artichoke as a vegetable rennet is that its greatest coagulating potential in milk is in the ripe flower, whereas artichoke cultivation does not focus on obtaining the ripe flower but rather on obtaining the immature flower, which will be consumed [5]. This reduces the availability of mature flowers for coagulant purposes. However, it should be noted that since artichoke is a high-demand crop, it is possible that there will be a greater availability of the mature flower compared to the availability of the mature thistle flower, which is not as widely cultivated, to the point that the FAO and its registry (FAOSTAT) do not present specific data on the cultivation of thistle at the world level [4].

## 7. Conclusions

Milk coagulation is strongly linked to the high enzymatic activity that affects к-casein and subsequently micelles. The protease enzymes present in most of the vegetable extracts used for milk coagulation are those responsible for the mentioned enzymatic activity.

The presence of protease enzymes in an extract of vegetable origin does not guarantee the formation of cheeses of excellent quality since vegetable rennets can confer undesirable organoleptic characteristics of the cheeses, probably due to the high proteolytic activity of these extracts. This phenomenon can be controlled by using milk with different levels of fat and protein than cow’s milk or by using a mixture of different types of milk, as in the case of PDO cheeses from the Iberian Peninsula, which are often made with sheep’s milk.

We advise carrying out further studies to attempt to reduce the aforementioned phenomenon related to proteolytic activity. Researchers should investigate methods to extract the coagulating potential of a plant using organs with lower coagulating activity, like in the artichoke (*Cynara scolymus*), or the purification of the protease enzymes present in the extracts since the high proteolytic activity is likely due to the diversity of enzymes in the extracts.

Cheeses aged less than 5 months made with vegetable rennets from the *Cynara* genus have similar organoleptic characteristics to cheeses made with traditional rennets but have varying levels of nitrogen. Therefore, it would be appropriate to improve the use of vegetable rennets by conducting further research on the ripening time, the presence of nitrogen in the cheeses, and how this may affect the shelf life of the cheeses and their sensory characteristics.

Due to the high availability of artichoke (*Cynara scolymus*) in the Mediterranean region, further research on methods of extraction and purification of its coagulant potential could make it an important industrial source of protease enzymes and high-value molecules such as inulin, improving the artichoke crop utilisation.

## Figures and Tables

**Figure 1 foods-12-03032-f001:**
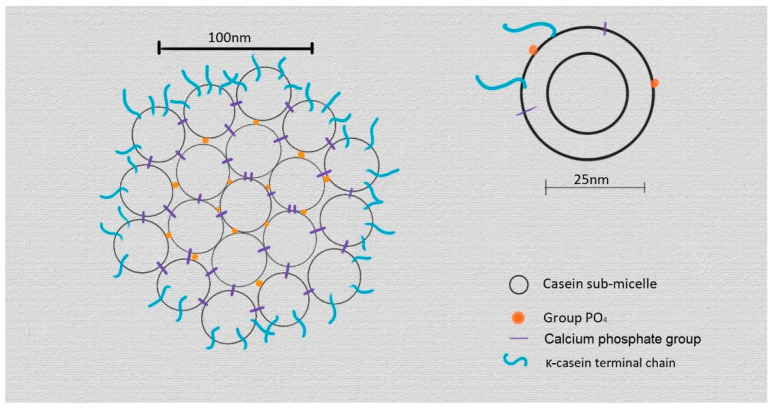
Representation of the micelle model proposed by Walstra and Jenness [23,24].

**Figure 2 foods-12-03032-f002:**
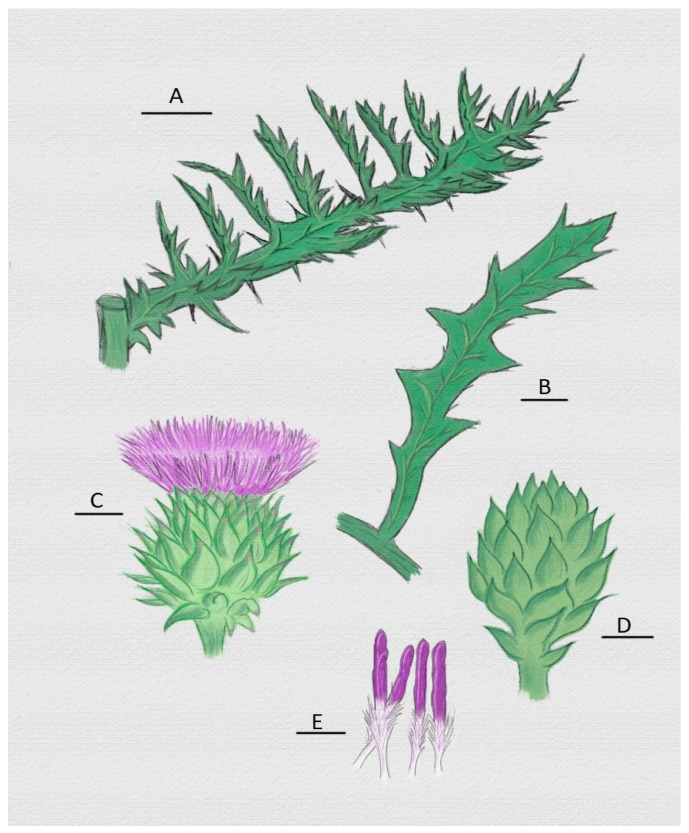
Representation of some organs of the cardoon (*Cynara cardunculus*) from which clotting enzyme can be extracted. (**A**) Mature leaf with thorns; (**B**) big leaf; (**C**) head and flower; (**D**) immature head; (**E**) pistils with their main parts: stylets and stigmas (white and purple part, respectively) [54,55].

**Figure 3 foods-12-03032-f003:**
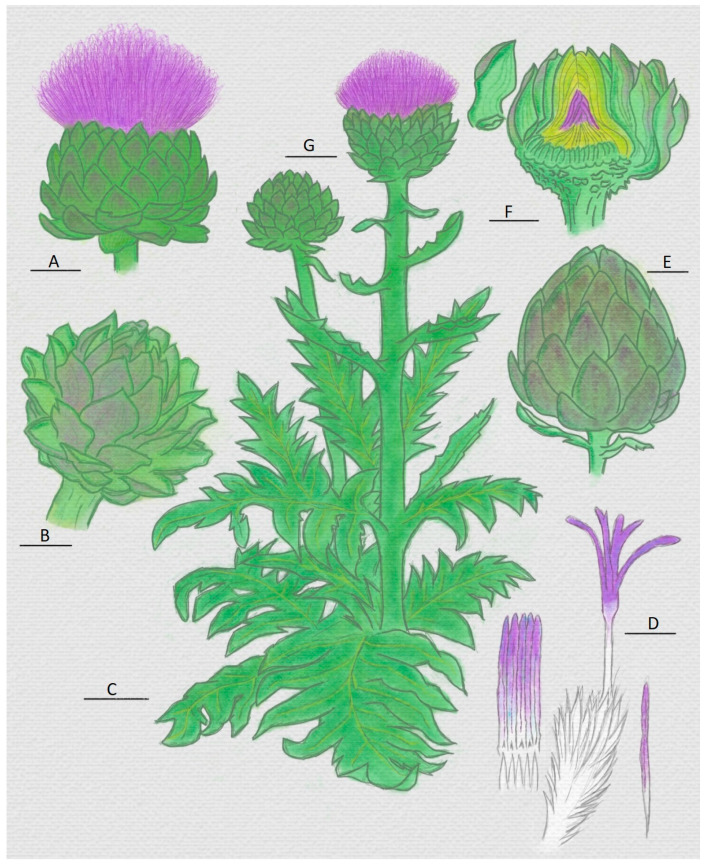
Representation of the artichoke (*Cynara scolymus*) plant and organs. (**A**) Head with flower; (**B**) head before maturation; (**C**) mature leaf; (**D**) pistils with their main parts: stylets and stigmas (white and purple part, respectively); (**E**) immature head; (**F**) head and internal heart; (**G**) stem and head with flower [55,66].

**Table 1 foods-12-03032-t001:** Reported values of milk’s general composition depending on animal species.

Animal/Component	Water %	Protein %	Fat %	Ash %	Lactose %	References
**Cow**	N/R	3.5	3–4	N/R	5	[9]
	N/R	3.2	3.6	0.7	4.7	[11]
	N/R	3.1–3.3	2.6–2.7	0.6–0.8	4.8–5.0	[12]
	85–87	3.2–3.8	3.7–4.4	0.7–0.8	4.8–4.9	[13]
**Buffalo/Yak**	83	4–6	6–9	N/R	5	[9]
	N/R	4.4–5.7	5.4–6.7	N/R	4.8–5.5	[14]
	82–84	3.3–3.6	7.0–11.5	0.8–0.9	4.5–5	[13]
	N/R	4.3–5.5	6.5–9.5	0.7–0.8	4.5–5.15	[15]
**Sheep**	N/R	5	5	N/R	6	[9]
	79–82	5.6–6.7	6.9–8.6	0.9–1.0	4.3–4.8	[13]
	N/R	5.42–6.86	6.0–8.4	-	4.32–5.18	[16]
	N/R	6.2	7.9	0.9	4.9	[11]
**Goat**	N/R	3.5	2–5	N/R	5	[9]
	N/R	3.4	3.8	0.8	4.1	[11]
	87–88	2.9–3.7	4.0–4.5	0.8–0.9	3.6–4.2	[13]
	N/R	3.5	3.8	0.8	4.1	[17]
**Camel**	N/R	4	3–4	N/R	5	[9]
	86–88	3.0–3.9	2.9–5.4	0.6–0.9	3.3	[13]
	N/R	2.7–2.9	2.6–2.9	1.1–1.3	4.6–4.8	[12]
	81.4–87	3.03–3.28	5.94–6.67	N/R	2.77–3.12	[18]

**Table 2 foods-12-03032-t002:** Types and sources of milk-clotting plant proteases.

Type of Protease	Protease Name	Source	References
**Aspartic**	Cardosins and cyprosins	*Cynara cardunculus*	[2,30,37]
	Cynarase	*C. scolymus*	[38,39,40]
	Procirsin	*Cirsium vulgare*	[30]
	Oryzasin	*Oryza sativa*	[30]
**Cysteine**	Ficin	*Ficus racemosa*	[30,41]
	Caprifig coagulant	*Ficus carica sylvestris*	[30]
	Actinidin	*Actinidia chinensis*	[30,42]
**Serine**	Cucumisin	*Cucumis melo*	[30]
	Lettucine	*Lactuca sativa*	[30]
	Streblin	*Streblus asper*	[30]

**Table 3 foods-12-03032-t003:** Cheeses with PDO designation of origin that are made with vegetable rennet.

Type/Name	Country/Region	Type of Plant	References
Torta del Casar	Spain/Extremadura	*Cynara cardunculus*	[51]
Queso Flor de Guía	Spain/Canary Islands	*C. cardunculus*, *C. scolymus*	
Queso de la Serena	Spain/Extremadura	*C. cardunculus*	
Queijo da Beira Baxa	Portugal	*C. cardunculus*	[52]
Queijo de Azeitão	Portugal	*C. cardunculus* L.	
Queijo Évora	Portugal	*C. cardunculus* L.	
Queijo de Nisa	Portugal	*C. cardunculus* L.	
Queijo Serra da Estrela	Portugal	*C. cardunculus*	

## Data Availability

The datasets generated for this study are available on request to the corresponding author.

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
