# Peer review of "Future Perspective and Technological Innovation in Cheese Making Using Artichoke (Cynara scolymus) as Vegetable Rennet: A Review"

_foods, 2023, doi:10.3390/foods12163032_

Round 1
Reviewer 1 Report
This paper suitable for publication after grammar editing.
This paper suitable for publication after grammar editing.
Reviewer 2 Report
Manuscript is good but some error are there.

ok
Reviewer 3 Report
In my opinion this article deals with a relevant topic for research, however it is not well written, since the text is confusing. Below are some comments:
The abstract does not present the main results found in the research.
The authors do not inform the main objective of the work, in general, it should be inserted at the end of the introduction.
The citation of the references in the text is still wrong, since, for example, in the text where it is written "Camila Soares and her working group..." and "Kethireddipalli and his team.." should be "Camila Soares [13] and her working group..." and "Kethireddipalli [30] and his team..."
The authors do not describe the methodology used, and it is simply impossible to identify how the research was carried out. In general, review articles must present the platform used, keywords and period to which the research refers.
The item “Future perspective for the use of artichoke” does not present a future perspective for this product.
There is no guarantee that the references used are sufficient for this type of review article and for this reason in my opinion it should be rejected.
The English language needs few revisions.
Round 2
Reviewer 3 Report
Dear Authors,
I appreciate all responses to all questions raised in this review, especially the fact that the methodology used to carry out this work was included.
It was impossible to publish this type of review article without a properly presented methodology.
Thanks!